# SUPPORT-SET BOTTLENECKS FOR VIDEO-TEXT REPRESENTATION LEARNING

**Mandela Patrick,** * **Po-Yao Huang,** * **Florian Metze & Andrea Vedaldi**
Facebook AI
{mandelapatrick,berniehuang,fmetze,vedaldi}@fb.com

**Alexander Hauptmann**
Language Technologies Institute
Carnegie Mellon University
alex@cs.cmu.edu

**Yuki M. Asano** *& **João Henriques**
Visual Geometry Group
University of Oxford
{yuki,joao}@robots.ox.ac.uk

## ABSTRACT

The dominant paradigm for learning video-text representations – noise contrastive learning – increases the similarity of the representations of pairs of samples that are known to be related, such as text and video from the same sample, and pushes away the representations of all other pairs. We posit that this last behaviour is too strict, enforcing dissimilar representations even for samples that are semantically-related – for example, visually similar videos or ones that share the same depicted action. In this paper, we propose a novel method that alleviates this by leveraging a generative model to naturally push these related samples together: each sample's caption must be reconstructed as a weighted combination of other support samples' visual representations. This simple idea ensures that representations are not overly-specialized to individual samples, are reusable across the dataset, and results in representations that explicitly encode semantics shared between samples, unlike noise contrastive learning. Our proposed method outperforms others by a large margin on MSR-VTT, VATEX, ActivityNet, and MSVD for video-to-text and text-to-video retrieval.

## 1 INTRODUCTION

Noise contrastive learning (Gutmann & Hyvärinen, 2010) is emerging as one of the best approaches to learn data representations both for supervised (Khosla et al., 2020) and unsupervised regimes (Chen et al., 2020c). The idea is to learn a representation that discriminates any two data samples while being invariant to certain data transformations. For example, one might learn a representation that identifies a specific image up to arbitrary rotations (Misra & van der Maaten, 2020). In a multi-modal setting, the transformations can separate different modalities, for example, by extracting the audio and visual signals from a video. The resulting noise contrastive representation associates audio and visual signals that come from the same source video, differentiating others (Patrick et al., 2020).

The noise contrastive approach is motivated by the fact that the transformations that are applied to the data samples leave their 'meaning' unchanged. For example, rotating an image does not change the fact that it contains a cat or not (Gidaris et al., 2018). However, in most cases, we expect to find many data samples that share the same content without being necessarily related by simple transformations (e.g. think of any two images of cats). Existing noise contrastive formulations are unaware of these relationships and still try to assign different representations to these samples (Wu et al., 2018), despite the fact that they are semantically equivalent. If the representation is learned for a downstream task such as semantic video retrieval, this might degrade performance.

This suggest that there might be other learning signals that could complement and improve pure contrastive formulations. In this paper, we explore this idea in the case of learning from two modali-

---

*Joint first authors.

Fig. 1: **Cross-modal discrimination and cross-captioning.** Our model learns from two complementary losses: (a) Cross-modal contrastive learning learns strong joint video-text embeddings, but every other sample is considered a negative, pushing away even semantically related captions (orange arrows). (b) We introduce a generative task of cross-captioning, which alleviates this by learning to reconstruct a sample's text representation as a weighted combination of a support-set, composed of video representations from other samples.

ties: videos and text, in the form of video transcripts or captions. Given a state-of-the-art contrastive formulation that learns from these two modalities, we investigate complementary pretext objectives to improve it. First, we consider the *(instance) captioning* task, namely mapping a video to the corresponding text, casting this as a conditional stochastic text generation problem. We show that this brings only a modest benefit.

We observe that the captioning task is highly sample-specific, as the goal is to produce a caption which describes a specific video and not any other video, and thus it suffers from the same disadvantages (discouraging concept sharing among samples) as contrastive learning. Thus, we propose to address this issue by switching to a different text generation task. The idea is to modify the text generator to take as input a learnable mixture of a support-set of videos, which we call *cross-instance captioning*. The mixture weights are generated by comparing the learned video representations to captions' representations in an online way over the batch. The limited set of support samples acts as a bottleneck that encourages extraction of shared semantics. In this manner, the embeddings can associate videos that share similar captions even if the contrastive loss tries to push them apart.

We show that, when the captioning task is added in this manner, it brings a sensible improvement to already very strong video representation learning results, further improving our own state-of-the-art baseline by a significant margin.

## 2 RELATED WORKS

Learning data representations from unlabelled data has been a long standing goal of machine learning. These approaches are called "self-supervised learning" because the learning signals, termed pretext tasks, are obtained from the data itself. In the image and video domain, pretext tasks include colorization (Zhang et al., 2016), rotation (Gidaris et al., 2018), or clustering (Asano et al., 2020a;b; Caron et al., 2018; Ji et al., 2018), while in the natural language domain, masked language modeling (Devlin et al., 2019), and next word prediction (Mikolov et al., 2013; Pennington et al., 2014) are extremely popular. These pretext tasks can be broadly classified into two classes: generative and discriminative.

Discriminative approaches learn representations by differentiating input samples, using objectives such as the contrastive loss (Gutmann & Hyvärinen, 2010; Hadsell et al., 2006). Discriminative approaches have proven to be particularly successful for image (Chen et al., 2020c; He et al., 2020; Misra & van der Maaten, 2020; Wu et al., 2018) and video (Han et al., 2019; Morgado et al., 2020; Patrick et al., 2020) representation learning. Generative approaches, on the other hand, try to reconstruct its input. GANs (Donahue & Simonyan, 2019; Goodfellow et al., 2014; Radford et al., 2015), autoencoders (Hinton & Salakhutdinov, 2006) and sequence-to-sequence models (Huang et al., 2020; Sutskever et al., 2014) are popular generative models. In this work, we show the importance of combining both discriminative and generative objectives to learn effective video-text representations.

The success of representation learning has also been due to advances in model architectures, such as the Transformer (Vaswani et al., 2017). BERT (Devlin et al., 2019) demonstrated that a transformer

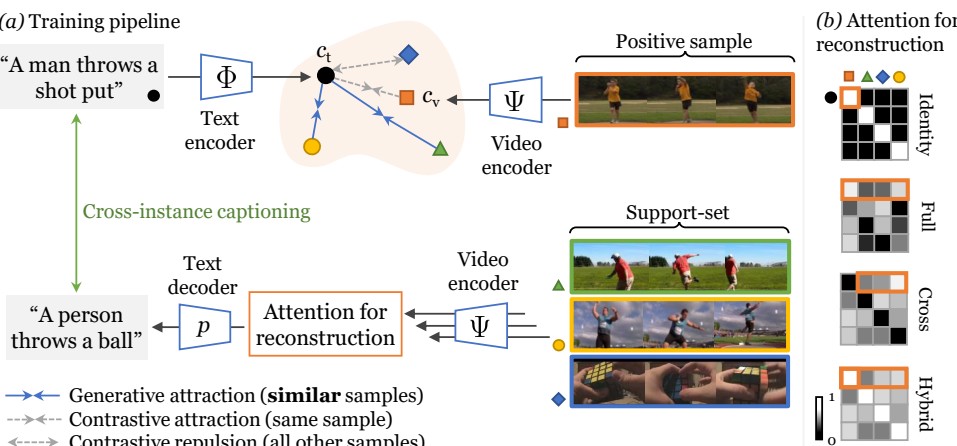

Fig. 2: **(a)** Our cross-modal framework with the discriminative (contrastive) objective and the generative objective. The model learns to associate video-text pairs in a common embedding space with text and video encoders (top). Meanwhile, the text must also be reconstructed as a weighted combination of video embeddings from a support-set (bottom), selected via attention, which enforces representation sharing between different samples. **(b)** Weights matrices (attention maps) used in each cross-captioning objective (see section 3.1.2).

architecture pretrained on large-scale textual data can learn transferable text representations that can be fine-tuned on a variety of downstream tasks. Subsequent works (Clark et al., 2020; Lewis et al., 2020a;b; Radford et al., 2019; Raffel et al., 2019) have improved upon the transformer architecture or training objective to learn even better representations. Inspired by the success of transformers in the NLP domain, several works have leveraged transformers to learn transferable image (Chen et al., 2020a; Desai & Johnson, 2020; Sariyildiz et al., 2020) or multi-modal image-text (Chen et al., 2019; Li et al., 2020a; 2019; Lu et al., 2019; Su et al., 2019; Tan & Bansal, 2019) and video-multilingual text (Huang et al., 2021) representations. In this work, we leverage the transformer architecture to better encode and represent text and video.

Large-scale training data has enabled the more effective pretraining of image (Sun et al., 2017; Yalniz et al., 2019), video (Ghadiyaram et al., 2019; Thomee et al., 2016) and textual representations (Raffel et al., 2019). The release of the HowTo100M dataset (Miech et al., 2019), a large-scale instructional video dataset, has spurred significant interest in leveraging large-scale pretraining to improve video-text representations for tasks such as video question-answering (Lei et al., 2018), text-video retrieval (Liu et al., 2019) and video captioning (Zhou et al., 2018b) on smaller datasets such as YouCookII (Zhou et al., 2018a), MSVD (Venugopalan et al., 2015a), MSR-VTT (Xu et al., 2016), LSMDC (Rohrbach et al., 2017), DiDeMo (Hendricks et al., 2018) and ActivityNet (Krishna et al., 2017). Although semantically rich and diverse, instructional videos from the web are super noisy and therefore a few approaches have been proposed to combat this. A few works (Luo et al., 2020; Sun et al., 2019a;b; Zhu & Yang, 2020) extend the BERT model to accept both visual and textual tokens to learn high-level semantic video-text representations. Other works have leveraged the contrastive loss (Miech et al., 2020) and show that using the raw audio (Alayrac et al., 2020; Rouditchenko et al., 2020) and other modalities (Gabeur et al., 2020) can be used to better align and improve video-text representations. While all these approaches rely on a contrastive objective, VidTranslate (Korbar et al., 2020) shows that a generative objective can also be used to learn joint video-text representations. In contrast to Korbar et al. (2020), we show that combining contrastive and generative objectives to pre-train video-text representations on large-scale data such as HowTo100M is very effective. The generative objective serves as regularizer to mitigate the strictness of the instance discrimination task of the contrastive objective, showing benefits similar to approaches such as clustering (Caron et al., 2020; Li et al., 2020b) and feature mixing (Kalantidis et al., 2020) which have been applied in the image domain.

## 3 METHOD

We consider the problem of learning multimodal representations from a corpus $\mathcal{C}$ of video-text pairs $(v, t)$, where $v$ is a video and $t$ is its corresponding text (caption or transcription). Our goal is to learn a pair of representation maps $c_v = \Psi(v)$ and $c_t = \Phi(t)$, with outputs in a $d$-dimensional embedding space $c_v, c_t \in \mathbb{R}^d$, where semantically similar instances are close to each other.

### 3.1 OBJECTIVE FOR LEARNING MULTIMODAL REPRESENTATIONS

We consider two learning objectives, also illustrated in Figure 1. The first is the contrastive objective, pushing embeddings $c_t$ and $c_v$ to be close if text $t$ and video $v$ come from the same sample and pushing them apart otherwise. This assumes that every sample is its own class and does not benefit from modelling similiarities *across* instances. The second objective is generative captioning. In its most basic variant, it maximizes the probability of generating the text $t$ given the corresponding video $v$. However, we suggest that variants that explicitly promote concept sharing between instances will result in better downstream performance, in tasks such as video retrieval. These variants, illustrated in Figure 2, have in common that the caption $t$ is reconstructed from a learned weighted combination over *other* videos $\hat{v}$. This is a form of attention (Bahdanau et al., 2014) which encourages the network to learn about which videos share similar semantics, compensating for the contrastive loss and grouping them implicitly.

In the following, we denote with $\mathcal{B} \subset \mathcal{C}$ a *batch* of multi-modal samples, i.e. a finite collection of video-text pairs $(t, v) \in \mathcal{C}$. For simplicity, we denote the batch as $\mathcal{B} = \{(t^i, v^i)\}_{i=1}^{B}\}$.

#### 3.1.1 CONTRASTIVE OBJECTIVE

To define the contrastive objective, let $s(a, b) = \frac{a^\top b}{\|a\|\|b\|}$ be the similarity measure between vectors $a$ and $b$. Following Faghri et al. (2018), we adopt the hinge-based triplet ranking loss with hard negative mining:

$$\mathcal{L}^{\text{contrast}} = \frac{1}{B} \sum_{i=1}^{B} \left[ \max_j \left[ \alpha - s(c_t^i, c_v^i) + s(c_t^i, c_v^j) \right]_+ + \max_j \left[ \alpha - s(c_t^i, c_v^i) + s(c_t^j, c_v^i) \right]_+ \right], \quad (1)$$

where $\alpha$ is the correlation margin between positive and negative pairs and $[\cdot]_+ = \max\{0, \cdot\}$ is the hinge function. In our experiments, we set $\alpha = 0.2$.

#### 3.1.2 CROSS-CAPTIONING OBJECTIVES

In the conventional captioning, the decoder seeks to optimize the negative log-likelihood of a text sequence $t$ given its corresponding video $v$:

$$\mathcal{L}^{\text{caption}} = -\frac{1}{B} \sum_{i=1}^{B} \log p(t^i | e_v^i). \quad (2)$$

Here, the log-likelihood is obtained via auto-regressive decoding (Vaswani et al., 2017) from an intermediate video embedding $e_v^i = \Phi'(v^i)$. For the cross-captioning objective, we modify this loss to condition the generation process on a weighted average of the embeddings of the *other* videos in the batch, which we call the *support-set*. The weights themselves, which can be interpreted as a batch-wise attention, are obtained as a softmax distribution with temperature $T$ over batch indices based on the video embeddings, as follows:

$$\mathcal{L}^{\text{cross-captioning}} = -\frac{1}{B} \sum_{i=1}^{B} \log p(t^i | \bar{e}_v^i), \quad \bar{e}_v^i = \sum_{j \in \mathcal{S}_i} \frac{\exp \langle c_t^i, c_v^j \rangle / T}{\sum_{k \in \mathcal{S}_i} \exp \langle c_t^i, c_v^k \rangle / T} \cdot e_v^j. \quad (3)$$

By default, the summation in the softmax is conducted over a support set $\mathcal{S}_i$ containing all indices except $i$. In the experiments, we consider the following attention types for reconstruction. **Identity captioning** ($\mathcal{S}_i = \{i\}$) generates the caption from the corresponding video and reduces to the standard captioning objective, eq. (2). **Full support** ($\mathcal{S}_i = \{1, \ldots, B\}$) considers all videos as possible candidates for captioning. **Hybrid captioning** sets the weights in eq. (3) as the average of the

weights for identity captioning and full support. **Cross-captioning** ($\mathcal{S}_i = \{j \neq i\}$) considers all *but* the video that one wishes to caption. This variant forces the network to extract all information required for captioning from other videos in the batch. Figure 2 compares graphically these attention mechanisms.

Considering both discriminative and generative objectives for learning multimodal representations, our full objective is $\mathcal{L} = \mathcal{L}^{\text{contrast}} + \lambda \mathcal{L}^{\text{cross-captioning}}$, where $\lambda$ balances two objectives. We set $\lambda = 10$ to ensure similar magnitudes for both losses in our experiments. In the training phase, we use Adam (Kingma & Ba, 2015) to minimize our loss. At inference time, we directly use $\Phi(t)$ and $\Psi(v)$ to encode video and text representations for retrieval.

## 3.2 MODEL ARCHITECTURE

We now discuss the details of the encoders and decoder components in our architecture, illustrated in fig. 2. For the *text decoder* $p(t|e_v)$ in eq. (2) and (3), we use a pre-trained T-5 decoder (Raffel et al., 2019).

For the *video representation* $c_v = \Psi(v) = \Psi''(\Psi'(v))$, we use a video encoder $e_v = \Psi'(v)$ followed by a multi-layer transformer pooling head $c_v = \Psi''(e_v)$. The encoder $\Psi'(v)$ concatenates the output of pretrained ResNet-152 (He et al., 2016) and R(2+1)D-34 (Tran et al., 2018) networks applied to individual video frames, resulting in a code $e_v = [e_{v1} \cdots e_{vM}]$ where $M$ is the maximum duration of a video clip. For the pooling head $c_v = \Psi''(e_v)$, we consider a transformer architecture to attend to important context and summarize it into a fixed-length representation $c_v$. For this, we follow MMT (Gabeur et al., 2020), but with two important differences. First, while MMT uses 7 expert features that results in $7\times$ the sequence length, we only use a transformer to attend to early-fused motion and appearance features as the video representation, thus significantly reducing the sequence length and computational cost. Second, instead of stacking 6 transformer layers to encode the visual stream as in MMT, we only use a shallow two-layer transformer architecture with additional pre-encoders, further increasing model efficiency. As temporal 1D-convolutional neural networks (CNNs) (LeCun et al., 1998) were shown to effectively capture temporal dependencies in videos (Dong et al., 2019), we integrate CNNs into our transformer pooling heads to better capture video temporal signals. In more detail, we compute $c_v = \Psi''(e_v)$ by chaining two transformer layers, each of the type:

$$\psi(e) = \text{BN}(\text{FFN}(e_{\text{attn}}) + e_{\text{attn}}), \quad e_{\text{attn}} = \text{BN}(\text{MHA}(f(e)) + f(e)). \tag{4}$$

Here $f$ is a pre-encoder that refines the video representation; we found empirically that a 1D CNN works well for this purpose. Then, we apply multi-head self-attention (MHA) (Huang et al., 2019; Vaswani et al., 2017) followed by a feed-forward network (FNN) with batch normalization (BN) (Ioffe & Szegedy, 2015). The architecture maps the input sequence $e_v$ to a new 'contextualized' sequence of representation vectors; we take the first one as $c_v$.

The text representation decomposes in the same way as $c_t = \Phi(t) = \Phi''(\Phi'(t))$. The text encoder $e_t = \Phi'(t)$ uses a pretrained T-5 network resulting in a code $e_t = [e_{t1} \cdots e_{tN}]$, where $N$ is the maximum length of a sentence. The pooling head $c_t = \Phi''(e_t)$ follows the same design as the video case, but $f$ is set to a recurrent neural network (RNN) instead of a CNN. Please refer to the appendix for details.

In practice, for computational reasons, we use eq. (3) to finetune the parameters of all networks except the video encoder $\Psi'(v)$, which is fixed.

## 4 EXPERIMENTS

We validate empirically the ability of our method to learn better representations for the downstream tasks of text-to-video and video-to-text retrieval. First, in sec. 4.2 we ablate various model components on the MSR-VTT dataset. Then, in sec. 4.3 we show that our best model significantly outperforms state-of-the-art retrieval systems on three datasets, MSR-VTT, ActivtyNet and VATEX. Finally, in sec. 4.4 we analyse qualitatively the effect of the attention mechanism used during training.

Table 2: **Model Architecture and Training Details Ablation.** Text→Video retrieval performance on MSR-VTT. Recall@1, 5, and Median Recall are shown.

(a) **Video Encoder**. Stronger features and combination improves performance.

| Feature source | $R@1\uparrow$ | $R@5\uparrow$ | $MdR\downarrow$ |
|---|---|---|---|
| R-152 | 20.8 | 46.2 | 6.0 |
| R(2+1)D-34 | 23.7 | 53.2 | 4.0 |
| R(2+1)D-34 +R-152 | **27.2** | **55.2** | **3.0** |

(b) **Feature Aggregation.** Learning temporal attention yields strong gains over pooling.

| Temporal reduction | $R@1\uparrow$ | $R@5\uparrow$ | $MdR\downarrow$ |
|---|---|---|---|
| Max | 21.8 | 49.5 | 8.0 |
| Mean | 22.5 | 51.3 | 6.0 |
| Multi-Head Attn | **27.2** | **55.2** | **3.0** |

(c) **Text Encoder.** Stronger encoding of text improves retrieval.

| Text Encoder | $R@1\uparrow$ | $R@5\uparrow$ | $MdR\downarrow$ |
|---|---|---|---|
| W2V (GloVe) | 22.1 | 49.8 | 6.0 |
| T5-Small | 24.5 | 51.2 | 3.0 |
| T5-Base | **27.2** | **55.2** | **3.0** |

(d) **Text Decoder.** Stronger decoding of text improves retrieval.

| Text Encoder | Text Decoder | $R@1\uparrow$ | $R@5\uparrow$ | $MdR\downarrow$ |
|---|---|---|---|---|
| T5-Base | T5-Small | 26.2 | 54.2 | 3.0 |
| T5-Base | T5-Base | **27.2** | **55.2** | **3.0** |

(e) **Contrastive Loss.** Inter-modal Triplet loss yields the best performance.

| Contrastive | $R@1\uparrow$ | $R@5\uparrow$ | $MdR\downarrow$ |
|---|---|---|---|
| InfoNCE (inter+intra) | 10.7 | 28.5 | 15.0 |
| InfoNCE (inter) | 10.8 | 29.0 | 14.5 |
| Triplet (inter+intra) | 26.8 | **56.2** | 3.0 |
| Triplet (inter) | **27.2** | 55.2 | **3.0** |

(f) **Support-set Size.** Retrieval degrades when reconstructing from too small and too large sets.

| Size | Batch-size | | | | | | | Memory bank | |
|---|---|---|---|---|---|---|---|---|---|
| | 8 | 16 | 32 | 64 | 128 | 256 | 512 | 2k | 8k |
| **R@1/5** | 18.5/45.6 | 20.7/49.9 | 25.2/54.6 | 27.2/55.2 | **28.0/56.1** | 26.9/55.0 | 25.3/53.5 | 26.8/54.7 | 26.2/52.7 |

## 4.1 EXPERIMENTAL SETUP

**Datasets.** **HowTo100M** (Miech et al., 2019) is a large-scale instructional video collection of 1.2 million YouTube videos, along with automatic speech recognition transcripts. We use this dataset for our pre-training experiments. **MSR-VTT** (Xu et al., 2016) contains 10,000 videos, where each video is annotated with 20 descriptions. We report results on the 1k-A split (9,000 training, 1,000 testing) as in Liu et al. (2019). **VATEX** (Wang et al., 2019) is a multilingual (Chinese and English) video-text dataset with 34,911 videos. We use the official training split with 25,991 videos and report on the validation split as in HGR (Chen et al., 2020b). The **ActivityNet Caption** (Krishna et al., 2017) dataset consists of densely annotated temporal segments of 20K YouTube videos. We use the 10K training split to train from scratch/ finetune the model and report the performance on the 5K 'val1' split. The **MSVD** (Chen & Dolan, 2011) dataset consists of $80K$ English descriptions for 1,970 videos from YouTube, with each video associated with around 40 sentences each. We use the standard split of 1,200, 100, and 670 videos for training, validation, and testing (Liu et al., 2019; Venugopalan et al., 2015b; Xu et al., 2015).

**Evaluation Metrics.** To measure the text-to-video and video-to-text retrieval performance, we choose Recall at K (R@K) and Median Rank (MedR), which are common metrics in information retrieval.

## 4.2 ABLATIONS

In Tab. 2, we first only ablate the cross-modal retrieval part of our network architecture, while the generative objectives are analysed in Tab. 1.

Table 1: **Effect of learning objectives.** Text→Video retrieval on MSR-VTT.

| | $R@1\uparrow$ | $R@5\uparrow$ | $MdR\downarrow$ |
|---|---|---|---|
| None | 25.9 | 53.0 | 4.0 |
| Identity | 26.4 | 51.9 | 4.0 |
| Full | 25.8 | 53.9 | 3.0 |
| Hybrid | 26.0 | 54.8 | 3.0 |
| Cross | **27.2** | **55.2** | **3.0** |

**Video Encoder.** In Tab. 2a, we show the effect of the choice of visual input features. We find that for text-to-video retrieval at Recall at 1 and 5 ($R@1, R@5$), features obtained from a video R(2+1)D-34 ResNet achieve $2.9\%$ and $7.0\%$ higher performance compared to only image-frame based features from a ResNet-152. A further $3.5\%$ and $2.0\%$ can be gained by concatenating both features, yielding the strongest $MdR$ of $3.0\%$.

**Feature Aggregation.** While the features from both video and image-based visual encoders have reduced spatial extent after a fully-connected layer, the temporal dimension can be reduced in various ways. In Tab. 2b, we find that our multi-head, parameterized attention reduction yields strong gains over the mean- or max-pooling baselines of over $4\%$ for $R@1$. This shows that learning attention over the temporal dimension of fixed feature sets can give strong gains even without fine-tuning the encoder.

**Text Encoder.** In Tab. 2c, we find decent gains of $2.7\%$ and $0.4\%$ for R@1,5 for using T5-base, instead of T5-small. We do not use the T-5-Large model, as in Korbar et al. (2020), due to the prohibitively large relative model size increase of +220%.

**Text Decoder.** In Tab. 2d, we find that using a larger text decoder gives a $1\%$ increase in performance when using the cross-captioning objective.

**Contrastive Loss.** To validate the choice of a triplet loss in eq. (1), in Tab. 2e, we compare the results of the InfoNCE contrastive loss (Oord et al., 2018) with a triplet loss, with both the intra and inter-intra modality variants. We find that InfoNCE (Oord et al., 2018) loss does not work well in our case, likely due to the difficulty in tuning this loss to have the right combination of temperature and batch-size.

**Support-Set Size.** Lastly, in Tab. 2f, we show the effect of the size of the support set used for cross-instance captioning. We find that our reconstruction loss indeed acts as a bottleneck, with both smaller and very large sizes degrading the performance.

**Captioning Objective.** In Tab. 1, we show the effect of the different variants of our learning objective eq. (3). First, we find that the naive addition of a reconstruction objective ("Identity") does not improve the contrastive-only baseline ("None") much. Considering reconstruction from other videos improves the performance more. In particular, the "Hybrid" variant, which combines "Identity" and "Full" (sec. 3.1.2) improves Recall at 1 and 5 from $25.9\%$ and $53.0\%$ to $26.0\%$ and $54.8\%$, respectively. However, the best result by far ($27.2/55.2\%$) is obtained forcing captions to be reconstructed only from *other* videos, via our cross-instance attention mechanism ("Cross"). This variant cannot use information contained in a video to generate the corresponding caption and thus entirely relies on the model to discover meaningful relationship between different videos. This newly-proposed scheme seems to have the most beneficial effect for semantic retrieval.

Table 3: **Retrieval performance on the MSR-VTT dataset.** Models in the second group are additionally pretrained on HowTo100M.

| | **Text →Video** | | | | **Video →Text** | | | |
|---|---|---|---|---|---|---|---|---|
| | $R@1$↑ | $R@5$↑ | $R@10$↑ | MdR↓ | $R@1$↑ | $R@5$↑ | $R@10$↑ | MdR↓ |
| Random Baseline | 0.1 | 0.5 | 1.0 | 500.0 | 0.1 | 0.5 | 1.0 | 500.0 |
| JSFusion (Yu et al., 2018) | 10.2 | 31.2 | 43.2 | 13.0 | – | – | – | – |
| HT100M (Miech et al., 2019) | 12.1 | 35.0 | 48.0 | 12.0 | – | – | – | – |
| JPoSE (Wray et al., 2019) | 14.3 | 38.1 | 53.0 | 9.0 | 16.4 | 41.3 | 54.4 | 8.7 |
| CE (Liu et al., 2019) | 20.9 | 48.8 | 62.4 | 6.0 | 20.6 | 50.3 | 64.0 | 5.3 |
| MMT (Gabeur et al., 2020) | 24.6 | 54.0 | 67.1 | 4.0 | 24.4 | 56.0 | 67.8 | 4.0 |
| **Ours** | **27.4** | **56.3** | **67.7** | **3.0** | **26.6** | **55.1** | **67.5** | **3.0** |
| VidTranslate (Korbar et al., 2020) | 14.7 | – | 52.8 | – | – | – | – | – |
| HT100M (Miech et al., 2019) | 14.9 | 40.2 | 52.8 | 9.0 | 16.8 | 41.7 | 55.1 | 8.0 |
| NoiseEstimation (Amrani et al., 2020) | 17.4 | 41.6 | 53.6 | 8.0 | – | – | – | – |
| UniVL (Luo et al., 2020) | 21.2 | 49.6 | 63.1 | 6.0 | – | – | – | – |
| AVLnet (Rouditchenko et al., 2020) | 27.1 | 55.6 | 66.6 | 4.0 | 28.5 | 54.6 | 65.2 | 4.0 |
| MMT (Gabeur et al., 2020) | 26.6 | 57.1 | **69.6** | 4.0 | 27.0 | 57.5 | 69.7 | 3.7 |
| **Ours-pretrained** | **30.1** | **58.5** | 69.3 | **3.0** | **28.5** | **58.6** | **71.6** | **3.0** |

Table 4: **Retrieval performance on the VATEX dataset**

| | Text →Video | | | | Video →Text | | | |
|---|---|---|---|---|---|---|---|---|
| | $R@1\uparrow$ | $R@5\uparrow$ | $R@10\uparrow$ | $\text{Md}R\downarrow$ | $R@1\uparrow$ | $R@5\uparrow$ | $R@10\uparrow$ | $\text{Md}R\downarrow$ |
| Random Baseline | 0.2 | 0.7 | 1.05 | 2000.5 | 0.02 | 0.1 | 1.02 | 2100.5 |
| VSE (Kiros et al., 2014) | 28.0 | 64.3 | 76.9 | 3.0 | – | – | – | – |
| VSE++ (Faghri et al., 2018) | 33.7 | 70.1 | 81.0 | 2.0 | – | – | – | – |
| Dual (Dong et al., 2019) | 31.1 | 67.4 | 78.9 | 3.0 | – | – | – | – |
| HGR (Chen et al., 2020b) | 35.1 | 73.5 | 83.5 | 2.0 | – | – | – | – |
| **Ours** | **44.6** | **81.8** | **89.5** | **1.0** | **58.1** | **83.8** | **90.9** | **1.0** |
| **Ours-pretrained** | **45.9** | **82.4** | **90.4** | **1.0** | **61.2** | **85.2** | **91.8** | **1.0** |

Table 5: **Retrieval performance on ActivityNet**

| | Text →Video | | | | Video →Text | | | |
|---|---|---|---|---|---|---|---|---|
| | $R@1\uparrow$ | $R@5\uparrow$ | $R@50\uparrow$ | $\text{Md}R\downarrow$ | $R@1\uparrow$ | $R@5\uparrow$ | $R@50\uparrow$ | $\text{Md}R\downarrow$ |
| Random Baseline | 0.02 | 0.1 | 1.02 | 2458 | 0.02 | 0.1 | 1.02 | 2458 |
| FSE(Zhang et al., 2018) | 18.2 | 44.8 | 89.1 | 7.0 | 16.7 | 43.1 | 88.4 | 7.0 |
| CE (Liu et al., 2019) | 18.2 | 47.7 | 91.4 | 6.0 | 17.7 | 46.6 | 90.9 | 6.0 |
| HSE (Zhang et al., 2018) | 20.5 | 49.3 | – | – | 18.7 | 48.1 | – | – |
| MMT (Gabeur et al., 2020) | 22.7 | 54.2 | 93.2 | 5.0 | 22.9 | 54.8 | 93.1 | 4.3 |
| **Ours** | **26.8** | **58.1** | **93.5** | **3.0** | **25.5** | **57.3** | **93.5** | **3.0** |
| MMT-pretrained (Gabeur et al., 2020) | 28.7 | 61.4 | 94.5 | 3.3 | **28.9** | **61.1** | 94.3 | 4.0 |
| **Ours-pretrained** | **29.2** | **61.6** | **94.7** | **3.0** | 28.7 | 60.8 | **94.8** | **2.0** |

Table 6: **Retrieval performance on the MSVD dataset**

| | Text →Video | | | | Video →Text | | | |
|---|---|---|---|---|---|---|---|---|
| | $R@1\uparrow$ | $R@5\uparrow$ | $R@10\uparrow$ | $\text{Md}R\downarrow$ | $R@1\uparrow$ | $R@5\uparrow$ | $R@10\uparrow$ | $\text{Md}R\downarrow$ |
| VSE (Kiros et al., 2014) | 12.3 | 30.1 | 42.3 | 14.0 | – | – | – | – |
| VSE++ (Faghri et al., 2018) | 15.4 | 39.6 | 53.0 | 9.0 | – | – | – | – |
| Multi. Cues (Mithun et al., 2018) | 20.3 | 47.8 | 61.1 | 6.0 | – | – | – | – |
| CE (Liu et al., 2019) | 19.8 | 49.0 | 63.8 | 6.0 | – | – | – | – |
| **Ours** | **23.0** | **52.8** | **65.8** | **5.0** | **27.3** | **50.7** | **60.8** | **5.0** |
| **Ours-pretrained** | **28.4** | **60.0** | **72.9** | **4.0** | **34.7** | **59.9** | **70.0** | **3.0** |

### 4.3 COMPARISON TO STATE-OF-THE-ART

In this section, we compare the results of our method to other recent text-to-video and video-to-text retrieval approaches on various datasets. In Tab. 3 to 5, we show the results of our model applied to text-to-video and video-to-text retrieval on MSR-VTT, VATEX, ActivityNet and MSVD with and without pre-trainig on HowTo100M. Without pre-training, our method outperforms all others in all metrics and datasets. In particular, for the VATEX dataset, our retrieval performance at recall at 1 and 5 is $45.9\%$ and $82.4\%$, exceeding recent state-of-the-art methods (Chen et al., 2020b) by a margin of $9\%$. For ActivityNet, our model outperforms MMT by a margin of $4\%$ at recall at 1. With pre-training on HowTo100M, our performance further increases across the board. Notably, unlike MMT which uses 7 features, our model uses only 2 features and achieves state-of-the-art in most metrics.

### 4.4 ANALYSIS

In order to better understand the effect of our learning objective, we visualize the soft attention of our best-performing cross-instance reconstruction model in fig. 3. As we can see in the top-left square, which shows the pairwise attention between all pairs of videos in the batch, it is highly focused, with the model mostly attending one or two other instances in the batch.

For the first video's caption reconstruction (second row), we find that the model solely attends to another musical performance video that is in the batch, ignoring the others. For the second video (third row), the model focuses on another sample that shows the sea but differs in most other aspects since there are no semantically-equivalent clips in the batch. The third video shares a similar scenario. These examples show that the bottleneck is effective at forcing the model to avoid memorising the video-caption association of each clip in isolation, and attempt to match other clips more broadly, since an exact (or very close) match is not guaranteed.

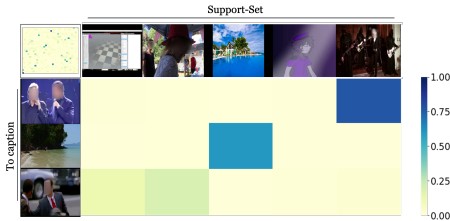

Fig. 3: **Support-set attention map.** Attention scores of all pairs in a batch (top-left square) and a subset of rows/columns (other squares) on VTT.

## 5 CONCLUSION

In this work, we studied classic contrastive learning methods such as the triplet loss to learn video-text representations for cross-model retrieval. We suggested that the contrastive approach might pull apart videos and captions even when they are semantically equivalent, which can hinder downstream retrieval performance. To mitigate this effect, we propose to consider a captioning pretext task as an additional learning objective. In particular, we show that cross-instance captioning can encourage the representation to pull together videos that share a similar caption, and are thus likely to be equivalent for retrieval. Leveraging these ideas, our model achieves state-of-the-art performance on the text-to-video and video-to-text retrieval tasks, on three datasets.

While we demonstrated these ideas in the specific case of text-to-video retrieval, they can in principle generalize to any setting that utilizes a contrastive loss, including self-supervised learning, provided that it is possible to learn reasonable conditional generators of a modality or data stream given another.

### ACKNOWLEDGEMENTS

We are grateful for support from the Rhodes Trust (M.P.), the Royal Academy of Engineering (DFR05420, J.H), Facebook (M.P. and P.H.), EPSRC Centre for Doctoral Training in Autonomous Intelligent Machines & Systems [EP/L015897/1] (M.P. and Y.A.) and the Qualcomm Innovation Fellowship (Y.A.). P.H. is also supported by the DARPA grant funded under the GAILA program (award HR00111990063).

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

## 6    APPENDIX

The appendix is organized as follows: First, we provide more details about our model. Then we introduce the datasets and the experimental setup. Finally, we provide additional qualitative and quantitative experimental results for video-text retrieval and captioning.

### 6.1    MODEL DETAILS

**Implementation details and hyper parameters.**    For our text encoder, we use the T5-base model pre-trained on the "Colossal Clean Crawled Corpus" (C4) (Raffel et al., 2019). We use its corresponding text tokenizer and encode a sentence into a sequence of 1024 dimensional vectors.

For our visual encoder, our model utilizes only the motion and the appearance features. For the motion feature, we use a 34-layer, R(2+1)-D (Tran et al., 2018) model pre-trained on IG65M (Ghadiyaram et al., 2019) and apply a spatial-temporal average pooling over the last convolutonal layer, resulting in a 512-dimensional vector. For the appearance feature, we use the 2048-dimension flattened pool-5 layer of the standard ResNet152 (He et al., 2016) pre-trained on Imagenet (Deng et al., 2009). We extract features at a rate of 1 feature per second and simply concatenate the two features, resulting in a 2560-dimension visual input stream. Noteworthily, instead of using 9 and 7 different types of visual features as in CE (Liu et al., 2019) and MMT (Gabeur et al., 2020), we use only the above 2 features and achieve on par or superior performance. Also, with early fusion, our model does not suffer from additional computation required for the extended sequence length in MMT. For the text decoder, we use the T5-base model decoder, also pre-trained on C4.

As illustrated in Fig. 4, our transformer pooling head is composed of a pre-encoder, a multi-head self-attention (MHA), and a feed-forward layer (FFN). For pre-encoders, we use a one-layer MLP with a $d$-dimensional output for mapping video features into the common embedding space. We use 1024-dimension bi-directional GRU as the text pre-encoder. For the 1D-CNN prior, we use kernels with size $[2, 3, 4, 6]$ as the visual and text pre-encoders. We set the embedding dimension to 1024 and use 4 attention heads in the transformer pooling layers. The hidden dimension of FFN is 2048.

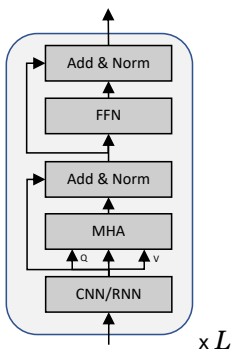

**Training and Inference time.**    Pre-training on 1.2 million HowTo100M videos takes around 160 GPU hours (NVIDIA V100) for 20 epochs. We speed up the pre-training process by distributing the workload over 8 GPUs. We use 1 GPU for the fine-tuning or training from scratch experiments. For the MSR-VTT 1k-A split, it takes 12 GPU hours to train our full model on 180K video-text pairs for 20 epochs. For Vatex, it takes 32 GPU hours to train on 260K video-text pairs for 30 epochs. For ActivityNet, it takes 2.5 GPU hours to train on 10K video-text paris for 28 epochs.

Fig. 4:  Transformer pooling head.

For inference, the encoding speed is around 250-300 video/sec and 200-250 text query/sec. The overall text-to-video search speed on 5,000 video-text pairs (5,000 text queries over 5,000 videos) is 30-34 seconds including encoding. The speed of text-to-video retrieval is similar to video-to-text retrieval.

### 6.2    EXPERIMENT DETAILS

The margin $\alpha$ of the max-margin loss is 0.2, and the temperature T is set to 0.1 as used in Sim-CLR Chen et al. (2020c). We use the Adam (Kingma & Ba, 2015) optimizer with a initial learning rate $5 \cdot 10^{-5}$ and clip gradients greater than 0.2 during the training phase. Dropout rate is 0.3 for all datasets besides ActivityNet (0.0).

As the average video/text lengths and videos available are quite different across datasets, we adjust our training scheme accordingly. When training on MSR-VTT, ActivtyNet and Vatex, batch-size is set to 64. For MSR-VTT training, we sample and truncate videos to 32 seconds, text to 100 tokens and train for 20 epochs. For Vatex, videos are at most 64 seconds and we train for 30 epochs. For ActivtityNet training, videos are at most 512 seconds and 256 tokens for the text part. We train

for 28 epochs on ActivityNet. For fine-tuning HowTo100M pre-trained model, we reduce training epochs into quarters.

## 6.3 DATASET DETAILS

**HowTo100M** (Miech et al., 2019) is a large-scale instructional video collection of 1.2 million Youtube videos, along with automatic speech recognition transcripts. There are more than 100 million clips (ASR segments) defined in HowTo100M. We use this dataset for pretraining.

**MSR-VTT** (Xu et al., 2016) contains 10,000 videos, where each video is annotated with 20 descriptions. For retrieval experiments and ablation studies, we follow the training protocol and defined in Gabeur et al. (2020); Liu et al. (2019); Miech et al. (2019) and evaluate on text-to-video and video-to-text search tasks on the 1k-A testing split with 1,000 video or text candidates defined by Yu et al. (2018). For captioning task, we evaluate on the standard testing split with 2,990 videos.

**VATEX** (Wang et al., 2019) is a multilingual (Chinese and English) video-text dataset with 34,911 videos. We use the official split with 25,991 videos for training. As the testing annotations are private in VATEX, we follow the protocol in Chen et al. (2020b) to split the validation set equally (1,500 validation and 1,500 testing videos) for model selection and testing. For each video, 10 English and 10 Chinese descriptions are available, and we only use the English annotations.

**ActivityNet Dense Caption** dataset consists densely annotated temporal segments of 20K YouTube videos. Following Gabeur et al. (2020); Zhang et al. (2018), we concatenate descriptions of segments in a video to construct "video-paragraph" for retrieval and captioning. We use the 10K training split to train from scratch/ finetune the model and report the performance on the 5K 'val1' split.

**MSVD** dataset consists of $80K$ English descriptions for 1,970 videos from YouTube, with each video associated with around 40 sentences each. We use the standard split of 1200, 100, and 670 videos for training, validation, and testing (Liu et al., 2019; Venugopalan et al., 2015b; Xu et al., 2015).

## 6.4 VIDEO CAPTIONING EXPERIMENTS

To measure captioning/text generation performance, we report BLEU4 (Papineni et al., 2002), METEOR (Denkowski & Lavie, 2014), Rogue-L (Lin, 2004) and CIDEr (Vedantam et al., 2015) metrics. We report results on the MSR-VTT, VATEX and ActivityNet datasets.

Table 7: **Captioning performance on the MSR-VTT dataset**

|  | Captioning | | | |
| --- | --- | --- | --- | --- |
|  | BLUE4 | METEOR | Rogue-L | CIDEr |
| VidTranslate (Korbar et al., 2020) | 41.7 | 28.5 | – | – |
| POS+VCT (Hou et al., 2019) | 42.3 | 29.7 | 62.8 | 49.1 |
| ORG (Zhang et al., 2020) | 43.6 | 28.8 | 62.1 | 50.9 |
| **Ours, MSR-VTT only** | 39.7 | 28.3 | 60.5 | 46.5 |
| **Ours, HT100M + MSR-VTT** | 38.9 | 28.2 | 59.8 | 48.6 |

Table 8: **Captioning performance on the VATEX dataset**

|  | Captioning | | | |
| --- | --- | --- | --- | --- |
|  | Blue@4 | METEOR | Rogue-L | CIDEr |
| Shared Enc-Dec (Wang et al., 2019) | 28.4 | 21.7 | 47.0 | 45.1 |
| ORG (Zhang et al., 2020) | 32.1 | 22.2 | 48.9 | 49.7 |
| **Ours, VATEX only** | **32.8** | **24.4** | **49.1** | **51.2** |
| **Ours, HT100M + Vatex** | 32.5 | 24.1 | 48.9 | 50.5 |

Table 9: **Captioning performance on the ActivtyNet dataset**

| | Captioning | | | |
|---|---|---|---|---|
| | Blue@4 | METEOR | Rogue-L | CIDEr |
| DENSE (Krishna et al., 2017) | 1.6 | 8.9 | – | – |
| DVC-D-A (Li et al., 2018) | 1.7 | 9.3 | – | – |
| Bi-LSTM+TempoAttn (Zhou et al., 2018b) | 2.1 | 10.0 | – | – |
| Masked Transformer (Zhou et al., 2018b) | 2.8 | 11.1 | – | – |
| **Ours, ActivityNet only** | 1.5 | 6.9 | 17.8 | 3.2 |
| **Ours, HT100M + ActivityNet** | 1.4 | 6.9 | 17.5 | 3.1 |

## 6.5 ZERO-SHOT RETRIEVAL EXPERIMENTS

We also evaluate our model in the zero-shot setting on MSR-VTT, Vatex, ActivityNet and MSVD, after pre-training on HT100M. While we are able to get reasonable results on MSR-VTT and MSVD, our results are not great on Vatex and Activity-Net due to significant domain gap.

Table 10: **Zero-shot Retrieval performance on VATEX, MSR-VTT, MSVD and ActivityNet.**

| | Text $\rightarrow$Video | | | | Video $\rightarrow$Text | | | |
|---|---|---|---|---|---|---|---|---|
| | $R@1\uparrow$ | $R@5\uparrow$ | $R@10\uparrow$ | $MdR\downarrow$ | $R@1\uparrow$ | $R@5\uparrow$ | $R@10\uparrow$ | $MdR\downarrow$ |
| *Zero-Shot* | | | | | | | | |
| ActivityNet | 0.06 | 0.2 | 0.5 | 1907.0 | 0.0 | 0.2 | 0.3 | 2238.0 |
| VATEX | 0.07 | 0.4 | 0.7 | 682.0 | 0.07 | 0.4 | 0.9 | 697 |
| MSVD | 8.9 | 26.0 | 37.9 | 18.0 | 21.4 | 46.2 | 57.7 | 6.0 |
| MSR-VTT | 8.7 | 23.0 | 31.1 | 31.0 | 12.7 | 27.5 | 36.2 | 24.0 |

## 6.6 ACTION RECOGNITION

Lastly, we evaluate our model on the video action recognition task on HMDB-51 (Kuehne et al., 2011) and UCF-101 (Soomro et al., 2012). For this, we use the R(2+1)D-34 (pretrained on IG65M) model as well as a ResNet-152 model (pretrained on Imagenet), as in our method. We extract a feature per second per video by concatenating the features from each model (2560-D), and obtain an average representation per video using either average pooling (2560-D) or our proposed transformer pooling head (1024-D) pre-trained on HT100M using cross-captioning objective. We then train a linear classifier for 1500 epochs for HMDB-51 (500 for UCF-101) on these features using Adam (Kingma & Ba, 2015) optimizer with learning rate of $1e^{-4}$ and weight decay $1e^{-4}$ with early stopping. We also drop the learning rate by 10 at epochs 200, 400 for UCF-101 and 1000, 1200 for HMDB-51. In Table 11, we show the results of training only a linear-layer on features extracted from our fixed backbone with or without a learned transformer-pooling head. We find that our transformer temporal pooling head provides significant benefits over the baseline of simply average pooling the features, demonstrating the effectiveness of building contextualized representations using our proposed transformer. In particular, we see improvements of over $7\%$ on HMDB-51 and $34\%$ on UCF-101 by replacing average pooling with our transformer pooling head to aggregate features. We observe that naive average pooling performs significantly worse than our transformer pooling under evaluation protocol. This is likely because 1) the average pooling collapses temporal information, making the linear layer based classification difficult 2) compared to the transformer pooling, it does not benefit from large-scale pretraining on a wide variety of action videos of HT100M. We further compare very favorably to the current state-of-the-art approaches. In particular, we outperform all other approaches, both supervised and self-supervised, except the recently introduced Omni (Duan et al., 2020) which was finetuned on both UCF-101 and HMDB-51, while we only trained a linear classifier on extracted features. However, it should be noted that it is very difficult to fairly compare all these different approaches because they may use different modalities (images, RGB video, optical flow, audio, ASR outputs), pretraining datasets (Kinetics-400, HT100M, IG65M, Imagenet), architectures (S3D, I3D, R(2+1)D, R3D), pre-training (supervised, self-supervised) and downstream training (frozen, finetuned) strategies.

Table 11: **Action recognition.** Results of training only a linear-layer, on features extracted from our fixed backbone with or without a learned transformer-pooling head. We compare to the state-of-art supervised and self-supervised pretrainig methods on the HMDB-51 and UCF-101 action recognition task, for different downstream training protocols ("FT?" stands for finetuned). We report average Top-1 accuracy across all 3 folds. Dataset abbreviations: **A**udio**S**et, **H**MDB**51**, **H**owTo100**M**, **I**nstagram**65M**, **IM**agenet-1000, **K**inetics**400**, **O**mni**S**ource Images + Videos, **S**ports**1M**, **U**CF**101**, **Y**ou**T**ube**8M**. Other abbreviations: **V**ideo modality, **F**low modality, **I**mage modality, **A**udio modality, **T**ransformer pooling, **AV**erage pooling

| Method | Mod | Dataset | Model | FT? | H51 | U101 |
|---|---|---|---|---|---|---|
| *Self-Supervised Pre-training* | | | | | | |
| MIL-NCE (Miech et al., 2020) | V,T | HM | S3D-G | ✗ | 53.1 | 82.7 |
| MIL-NCE (Miech et al., 2020) | V,T | HM | S3D-G | ✓ | 61.0 | 91.3 |
| MMV (Alayrac et al., 2020) | V,T,A | HM+AS | TSM-50x2 | ✗ | 67.1 | 91.8 |
| ELo (Piergiovanni et al., 2020) | V,F,A | YT8M | R(2+1)D-50x3 | ✓ | 67.4 | 93.8 |
| XDC (Alwassel et al., 2020) | V,A | IG65M | R(2+1)D-18 | ✓ | 68.9 | 95.5 |
| GDT (Patrick et al., 2020) | V,A | IG65M | R(2+1)D-18 | ✓ | 72.8 | 95.2 |
| MMV (Alayrac et al., 2020) | V,T,A | HM+AS | TSM-50x2 | ✓ | 75.0 | 95.2 |
| *Supervised Pre-training* | | | | | | |
| P3D (Qiu et al., 2017) | V,I | S1M+IM | P3D | ✓ | – | 88.6 |
| TSN (Wang et al., 2018) | V,I | IM | TSN | ✓ | 69.4 | 94.2 |
| I3D (Carreira & Zisserman, 2017) | V,I | K400+IM | I3D | ✓ | 74.8 | 95.6 |
| R(2+1)D (Tran et al., 2018) | V | K400 | R(2+1)D-34 | ✓ | 74.5 | 96.8 |
| S3D-G (Xie et al., 2018) | V,I | K400+IM | S3D-G | ✓ | 75.9 | 96.8 |
| I3D (Carreira & Zisserman, 2017) | V,I | K400+IM | I3D | ✓ | 77.1 | 96.7 |
| R(2+1)D (Tran et al., 2018) | V | K400 | R(2+1)D-34 | ✓ | 76.4 | 95.5 |
| R(2+1)D (Tran et al., 2018) | V,F | K400 | R(2+1)D-34x2 | ✓ | 78.7 | 97.3 |
| Omni (Duan et al., 2020) | V,I | K400+OS | Slow-8x8-R101 | ✓ | 79.0 | 97.3 |
| I3D (Carreira & Zisserman, 2017) | V,F,I | K400+IM | I3Dx2 | ✓ | 80.7 | 98.0 |
| Omni (Duan et al., 2020) | V,F,I | K400+OS | Slow-8x8-R101x2 | ✓ | **83.8** | **98.6** |
| Ours (Avg-pooling) | V,I | IG65M+IM | R(2+1)D-34+R152 | ✗ | 73.7 | 64.3 |
| Ours (T-pooling) | V,I | HM+IG65M+IM | R(2+1)D-34+R152 | ✗ | 81.3 | 98.0 |

## 6.7 STATISTICAL SIGNIFICANCE

In Table 12, we show the results of finetuning our pretrained model for 3 times on the VATEX dataset. We find that the variance is quite low and our model consistently beats the state of the art.

Table 12: **Retrieval performance on the VATEX dataset**

| | Text $\to$ Video | | | | Video $\to$ Text | | | |
|---|---|---|---|---|---|---|---|---|
| | $R@1\uparrow$ | $R@5\uparrow$ | $R@10\uparrow$ | MdR↓ | $R@1\uparrow$ | $R@5\uparrow$ | $R@10\uparrow$ | MdR↓ |
| Random Baseline | 0.2 | 0.7 | 1.05 | 2000.5 | 0.02 | 0.1 | 1.02 | 2100.5 |
| VSE (Kiros et al., 2014) | 28.0 | 64.3 | 76.9 | 3.0 | – | – | – | – |
| VSE++ (Faghri et al., 2018) | 33.7 | 70.1 | 81.0 | 2.0 | – | – | – | – |
| Dual (Dong et al., 2019) | 31.1 | 67.4 | 78.9 | 3.0 | – | – | – | – |
| HGR (Chen et al., 2020b) | 35.1 | 73.5 | 83.5 | 2.0 | – | – | – | – |
| **Ours** | **44.9**$_{\pm0.2}$ | **82.1**$_{\pm0.2}$ | **89.7**$_{\pm0.2}$ | **1.0** | **58.4**$_{\pm0.1}$ | **84.4**$_{\pm0.2}$ | **91.0**$_{\pm0.3}$ | **1.0** |

## 6.8 ADDITIONAL QUALITATIVE RESULTS

We provide addition qualitative text-to-video retrieval results on MSR-VTT, VATEX, ActivityNet in Fig. 5. Given a text query, in most cases, our model successfully retrieves the correct videos marked in green.

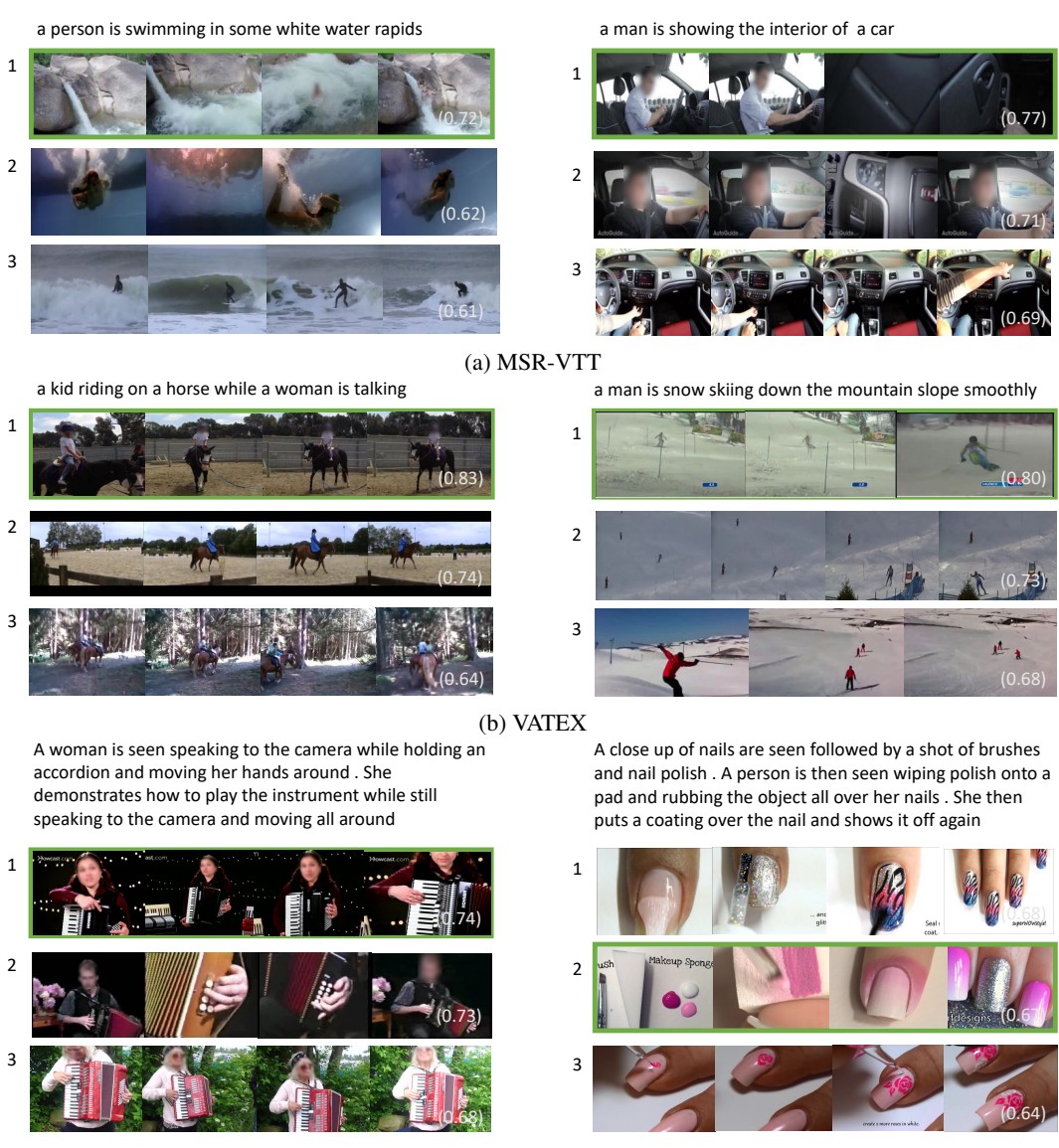

Fig. 5: Examples of top-3 Text→Video retrieval results and similarities on the MSR-VTT, VATEX, and ActivityNet testing set. Only one correct video (colored in green) for each text query on the top.

