# OpenReview forum: "Support-set bottlenecks for video-text representation learning"
_ICLR.cc/2021/Conference — ICLR 2021 Spotlight_

### Official Review · AnonReviewer4 · 2020-10-27
**An interesting objective for video-text models**

**Rating:** 7
**Confidence:** 5

**Review:**

## Summary

The paper proposes a new way to train joint representation of video and text. In particular, it argues that the contrastive based approach is too strict in the sense that it forces pairs of video and text to be far away only because they are not from the same instance without considering potential shared semantic. To address this, they propose to combine the contrastive loss with a generative loss that tries to generate the text from one video based only on *different* videos within the batch. Interestingly, adding this objective improves performance of retrieval.

## Strengths

* The paper is clearly written which makes it easy to understand
* To me the fact that the Cross approach works best is not completely intuitive which makes the result quite interesting. This approach is novel to the best of my knowledge.
* The ablation study is thorough and overall well conducted.
* The final performance are quite impressive (albeit some difficulty to fully compare to previous work as explained in the weaknesses section).

## Weaknesses

There are a few questions/concerns about the work that I hope can be answered in the rebuttal.

**Is the support set idea always good?**: While this idea seems to work well on MSRVTT I wonder whether or not the same loss would work well on much larger scale and more diverse datasets. In particular the method seems to rely strongly on the fact that there are other semantically related videos/text in the batch. While this may be true for small scale datasets it might not always be the case. I see a few options to try to illustrate a bit more that point:

* Is the support set idea also important when training on HowTo100M? Does it matter there? Does it hurt?
* I would expect the support set size to depend on the dataset? Was this observed in practice?

**What are zero shot performance when training on HowTo?** I would be curious to know what would be the ZS performance when just training on HowTo. In particular (related to the previous question) is the generation loss also important there at all?

**Comparison to the state-of-the-art is not totally fair**: even though less features are used, the pretrained models are trained on different data (IG65M dataset), therefore it's not clear that the comparison is fair compared to MMT. Would it be possible to do the comparison using the same set of features? Currently its not clear if the best performance comes from the fact that the method is better or if the input features are better.

**Technical questions about aggregation**:

* For the text encoder, it does not seem to be standard to combine RNN with a Transformer. Was it really important? Similarly for the vision model, how important was it to add the convolution in the transformer?

* Have you tried other options for $c_v$ (e.g. by averaging all the outputs instead of just taking the first? Asking since in that case it seems that no special [CLS] token was added so taking the first token looked arbitrary.

* When averaging temporal representations of the different videos in equation (3), this somewhat assume that all videos are temporally aligned. However this is often not true. Is this a problem that the authors have considered?


**Some details about the tables**:

* In Table 1. e) why the numbers with batch size 64 do not match the other best numbers in the table (55.7 vs 55.2).
* Table 1. d). What does inter+intra means? In particular for the intra does it mean you do a contrastive loss between video and video? Are there any augmentations that are used for that case? I might have missed this but I could not find the details of that in the paper.

**Comment about the title + question**: when I first read the title I thought that visual features would also be learned end to end (due to the term representation learning). I wonder if representation learning is the best term to put in the title due to the potential confusion. Related to that, have you tried to also finetune the video networks? If yes, were there benefits?

## Justification of rating

The paper presents a simple but effective idea for training joint text and video models. The paper is clearly written and the results are compelling and do support the claim of the paper. There are still some things that can be improved about the paper and I am expecting the authors to reply to my questions and concerns.

## Post rebuttal comment

The authors have successfully answered my concerns and added the necessary experiments. I think the paper should be accepted as the support set idea is an important one to improve upon the simple contrastive idea (in particular to soften the set of positives).

---

> ### Author Response · Authors · 2020-11-14
> **Response to Reviewer 4**
>
> Thank you R4 for your comments. We appreciate that you find our paper very clearly written, comprehensive and interesting, with impressive results.
>
> **Is the support set idea always good?: While this idea seems to work well on MSRVTT I wonder whether or not the same loss would work well on much larger scale and more diverse datasets. In particular the method seems to rely strongly on the fact that there are other semantically related videos/text in the batch. While this may be true for small scale datasets it might not always be the case. I see a few options to try to illustrate a bit more that point: Is the support set idea also important when training on HowTo100M? Does it matter there? Does it hurt?
> I would expect the support set size to depend on the dataset? Was this observed in practice?**
> While in the paper we already observe the support-set idea working well on a variety of datasets: VTT, VATEX (more diverse texts), ActivityNet (longer videos, and texts), we agree about the intuition.
> Indeed, the ideal support-set size might depend on the intrinsic diversity of the pretraining dataset and we will add an ablation in the updated manuscript showing the effect of cross-captioning when pretraining on HowTo100M.
>
> **What are zero shot performance when training on HowTo? I would be curious to know what would be the ZS performance when just training on HowTo. In particular (related to the previous question) is the generation loss also important there at all?**
> We will update the manuscript with zero-shot numbers as requested.
>
> **Comparison to the state-of-the-art is not totally fair: even though less features are used, the pretrained models are trained on different data (IG65M dataset), therefore it's not clear that the comparison is fair compared to MMT. Would it be possible to do the comparison using the same set of features? Currently its not clear if the best performance comes from the fact that the method is better or if the input features are better.**
> Compared to MMT, which uses 7 expert features (motion, image, face-detections, speech, OCR, scene, audio), we use only 2 features (motion and image) and achieve superior performance. Although we agree that it would be ideal to have an apples-to-apples comparison with MMT, the features they use for pretraining have not been publicly released, and we believe the results shown in the paper demonstrate the effectiveness of our proposed method convincingly.
>
> **For the text encoder, it does not seem to be standard to combine RNN with a Transformer. Was it really important? Similarly for the vision model, how important was it to add the convolution in the transformer?**
> As described in Sec 3.2, RNN+BERT embedding has been explored in previous works such as DualEncoder [1] and MIL-NCE [2]. CNNs have also been shown to be effective in DualEncoder as a proper N-gram-like prior. In our experiments we observed a 0.2~0.5 R@1 degradation in performance without them in our experiment.
>
> **When averaging temporal representations of the different videos in equation (3), this somewhat assume that all videos are temporally aligned. However this is often not true. Is this a problem that the authors have considered?**
> The video clips during training are sampled iid and we do not assume they are temporally aligned.
>
> **In Table 1. e) why the numbers with batch size 64 do not match the other best numbers in the table (55.7 vs 55.2).**
> Yes that was a typo, and we have updated the manuscript.
>
> **Table 1. d). What does inter+intra means? In particular for the intra does it mean you do a contrastive loss between video and video? Are there any augmentations that are used for that case? I might have missed this but I could not find the details of that in the paper.**
> Yea intra+inter means we combine both cross-modal (video, text) and within-modal losses (video, video). We use temporal jitter augmentations, where we sample two clips from the same video when computing the intra-modality loss. No other data augmentations were used.
>
> **Comment about the title + question: when I first read the title I thought that visual features would also be learned end to end (due to the term representation learning). I wonder if representation learning is the best term to put in the title due to the potential confusion. Related to that, have you tried to also finetune the video networks? If yes, were there benefits?**
> R(2+1)-D and T5 are both large models, so finetuning both would be very expensive computationally. Furthermore, in VidTranslate[3], they show that using a linear layer while fixing the video backbone (aka pre-extract visual features) is sufficient.
>
> **References**
> [1] Dong et al. Dual Encoding for Zero-Example Video Retrieval. CVPR 2019
> [2] Miech et al. End-to-End Learning of Visual Representations from Uncurated Instructional Videos. CVPR 2020.
> [3] Korbar et al. Video Understanding as Machine Translation. ArXiv 2020

---

### Official Review · AnonReviewer3 · 2020-10-28
**contrastive learning for video-text representation learning with a new proposed generative objective**

**Rating:** 6
**Confidence:** 4

**Review:**

Summary:
The paper introduces an interesting problem when doing noise contrastive learning. In particular, the model will push away the representations of all negative pairs although these samples sometimes are semantically related. To alleviate this problem, the authors develop a generative model to push these related samples together by reconstructing the caption from the weighted feature of support samples. Overall, the insight of semantically related pairs in negative sampling is interesting for me. My major concern is about the architecture of the proposed model, the performance on other tasks, and some additional ablation studies (see below).

Pros:
1. The paper takes an important issue in contractive learning: semantically related samples in negative sets.
2. The paper proposes the cross-instance captioning to alleviate the discouraging concept sharing among samples. The comprehensive experiments on retrieval tasks show SOTA results with a margin.

Cons:
1. A pretrained T5 with 12-layer encoder and 12-layer decoder is used in the model. It is doubt that why the cross-instance captioning needs such a heavy decoder. How about a shallow decoder affect the performance?
2. The caption results shown in the appendix demonstrate a weak advantage in generation tasks. Are there some reasons for the opposite performance compared with the retrieval tasks?
3. The model places the pretrained architectures after the initialized Transformer pooling head, which will affect the pretrained weights. What is the training strategy and learning rate to deal with such a problem?
4. What is the possible reason for the reconstruction bottleneck that both smaller and very large sizes degrading the performance? Besides, the performance is sensitive and high-impact with different sizes.
5. There is a doubt that the MdR of T5-Small in Table 1 (c) is the same as that of T5-Base but R@1 and R@5 have a margin.
6. What is the meaning of `None` in Table 2? Does it mean training without cross-instance captioning?
7. This paper sets the $\alpha$ to 0.2 in Eq. (1). How about other numbers affect the results, e.g., 0.1 and 0.3? What is the value of temperature T in Eq. (3)?
8. Are there some zero-shot results for the pretrained model?
9. How about the performance of the video representation on other tasks, e.g., action recognition?
10. The support-set is the samples of the mini-batch. How to know which one semantically relates to the anchor?

11. Missing references:
1). Linchao Zhu and Yi Yang. 2020. Actbert: Learning global-local video-text representations. In CVPR.
2). Huaishao Luo, Lei Ji, Botian Shi, Haoyang Huang, Nan Duan, Tianrui Li, Jason Li, Taroon Bharti, and Ming Zhou. 2020. UniVL: A Unified Video and Language Pre-Training Model for Multimodal Understanding and Generation. arXiv:2002.06353.
3). Bruno Korbar, Fabio Petroni, Rohit Girdhar, and Lorenzo Torresani. 2020. Video understanding as machine translation. arXiv:2006.07203.

---

> ### Author Response · Authors · 2020-11-14
> **Response to Reviewer 3**
>
> Thank you R3 for your detailed comments and feedback. We appreciate that you find our paper very comprehensive and tackling an important issue in contrastive learning.
>
> **A pretrained T5 with 12-layer encoder and 12-layer decoder is used in the model. It is doubt that why the cross-instance captioning needs such a heavy decoder. How about a shallow decoder affect the performance?**
> That is a good point. We are currently running this experiment and will update the manuscript (TO DO).
>
> **The caption results shown in the appendix demonstrate a weak advantage in generation tasks. Are there some reasons for the opposite performance compared with the retrieval tasks?**
> The main focus of our paper is video-text alignment for video-text retrieval tasks. As such, we add a generative task only as a form of regularisation and do not optimize for it. In addition, current state-of-the-art captioning performance can require many tricks such as using a self-critic [1] that directly optimizes BLEU/METEOR, or many data augmentation techniques, as well as using many visual features. We consider improving both alignment and text generation jointly as our next step.
>
> **The model places the pretrained architectures after the initialized Transformer pooling head, which will affect the pretrained weights. What is the training strategy and learning rate to deal with such a problem?**
> We used Transformer Pooling head after the pretrained backbone architecture, not before, as described in Sec 3.2.
>
> **What is the possible reason for the reconstruction bottleneck that both smaller and very large sizes degrading the performance? Besides, the performance is sensitive and high-impact with different sizes.**
> The reason for this U-shape in performance precisely validates our hypothesis of the additional generative task acting as a bottleneck: if the support-set is too small, the generative task cannot work properly, while if it is too large, it is too easy and does not regularize the training.
>
> **There is a doubt that the MdR of T5-Small in Table 1 (c) is the same as that of T5-Base but R@1 and R@5 have a margin.**
> We have double-checked and can confirm the accuracy of these values. Its because medR does not change as much and the recall scores are more sensitive.
>
> **What is the meaning of None in Table 2? Does it mean training without cross-instance captioning?**
> Correct: None refers to training with the baseline: ie. using a contrastive loss only.
>
> **This paper sets the $\alpha$ to 0.2 in Eq. (1). How about other numbers affect the results, e.g., 0.1 and 0.3? What is the value of temperature T in Eq. (3)?**
> For the $\alpha$ in triplet loss, we just use the setup as in VSE++ [2], same as in other video-text works. The value of temperature T is 0.1 as used in SimCLR [3]. Since both works have explored various settings for these parameters we simply adopt these.
>
> **Are there some zero-shot results for the pretrained model?**
> Yes, we will zero-shot numbers to our revised version.
>
> **How about the performance of the video representation on other tasks, e.g., action recognition?**
> While the focus of this work lies in obtaining strong video-text text-video retrieval methods, we agree that our model *does* learn visual feature representation due to the transformer pooling weights being learned.
> We will provide results on finetuning our visual encoder on two common video action recognition benchmarks HMDB-51 and UCF-101 in the updated version of our paper.
>
> **The support-set is the samples of the mini-batch. How to know which one semantically relates to the anchor?**
> The "relationship" is determined by closeness in the embedding space via our dot-product attention mechanism. We then generate our caption as weighted combination of the samples in the support-set. Please check Fig3. for a visualization of sematic closeness.
>
> **Missing references:
> Linchao Zhu and Yi Yang. 2020. Actbert: Learning global-local video-text representations. In CVPR. 2). Huaishao Luo, Lei Ji, Botian Shi, Haoyang Huang, Nan Duan, Tianrui Li, Jason Li, Taroon Bharti, and Ming Zhou. 2020. UniVL: A Unified Video and Language Pre-Training Model for Multimodal Understanding and Generation. arXiv:2002.06353. 3). Bruno Korbar, Fabio Petroni, Rohit Girdhar, and Lorenzo Torresani. 2020. Video understanding as machine translation. arXiv:2006.07203.**
> In our original submission, we have actually cited both ActBert, and VideoTranslate, but we thank the reviewer for the UniVL reference and will add its citation and their MSR-VTT results to the updated manuscript.
>
> **References**
> [1] Rennie et al. Self-critical Sequence Training for Image Captioning. CVPR 2017
> [2] Faghri et al. VSE++: Improving Visual-Semantic Embeddings with Hard Negatives. CVPR 2017.
> [3] Chen et al. A simple framework for contrastive learning of visual representations. ICML 2020

---

### Official Review · AnonReviewer1 · 2020-10-28
**Better learning of video-text representations via cross-captioning loss**

**Rating:** 9
**Confidence:** 3

**Review:**

Summary:
This paper focuses on better learning of video-text representations. To this end, the paper introduces a new generative task of cross-captioning which alleviates the typical issue of contrastive learning by learning to reconstruct a sample’s text representation as a weighted combination of a video support set. The proposed approach performs better than previous work on various datasets.

Pros:
1) The paper is well written and easy to follow.

2) The proposed method sounds novel and interesting. The empirical evaluations on various datasets suggest that the proposed method is better.

3) The ablations on various modules of the proposed method is very interesting and thorough.


Cons:
1) The current approach limits to using only the videos in the current batch for the support set. One could also try retrieving the support set from the full dataset in an online way. It would be interesting to see this ablation.

Overall:
The proposed method of cross-captioning is novel and the thorough empirical evaluations/ablations further show the superiority of the proposed method as well as the usefulness of each component.

Questions:
1) Please provide statistical significance scores wherever necessary, e.g., Table-4 ours vs ours-pretrained difference is statistically significant?

2) Is it possible to ablate on the choice of support set from within a batch vs. full dataset?

---

> ### Author Response · Authors · 2020-11-14
> **Response to Reviewer 1**
>
> Thank you R1 for your comments. We appreciate that you find our paper well written, easy to follow and novel. We answer both questions below:
>
> **Q1: Please provide statistical significance scores wherever necessary, e.g., Table-4 ours vs ours-pretrained difference is statistically significant?**
> Thank you for this suggestion, we agree that this will add more weight to our results. We will add statistical significance scores in our tables for a revised version of this paper.
>
> **Q2: Is it possible to ablate on the choice of support set from within a batch vs. full dataset?**
> We have a similar experiment already in the paper (Tab.1e): where instead of using a support set from the batch, we use a queue-based memory bank of size 2048 to leverage a larger set of samples. However, we found that it doesn't show significant improvements, and in fact hurts performance, validating our claim that the support-set acts as a bottleneck.
> To analyze whether this holds true for queue sizes approaching the full dataset, we will expand Table 1c. to show the effect of even larger sized memory banks (4k, 8k, note: the datasetsize ~10k).

---

### Official Review · AnonReviewer2 · 2020-10-28
**Strong empirical results; motivation needs clarification**

**Rating:** 7
**Confidence:** 4

**Review:**

This paper concerns the problem of video-text representation learning and its application to video-text retrieval. The source of pre-training data comes from YouTube video-ASR pairs. The main novelty is the adoption of a generative objective (i.e., video captioning) to refine the video encoder ($\Psi''$) and the text encoder, given the paired data input. Further, the paper observes that sometimes videos belong to the same batch could share similar characteristics/semantics, and therefore traditional contrastive learning objectives that repel positive samples from these hard negative samples could potentially hurt the learned representation. To this end, this paper introduces an idea called support-set bottlenecks, which effectively alleviate this conflict through batch-wise attention.

Like the review title indicated, the paper has demonstrated strong empirical results on some common benchmarks. The related work is mostly comprehensive. The method section is also described with good clarity. However, the main concerns include the somewhat weak motivation of the method and some overstated claims. The reasons are detailed as follows.

i) The motivation behind using a support set is still unclear, besides the obvious reason of empirical gain. Compare to related techniques such as semi-hard negative mining, what are their relations and the advantage of the proposed approach? Also, since the idea is generic to the training objective, what is the rationale for not using it for the contrastive objective (but instead on the more complicated captioning objective)? Or if you have made attempts, what are the observations?

ii) The overall scope of the proposed method is somewhat overstated. Despite that the method is generic to learn video-text representation, the downstream tasks only involve a single type of problem (i.e., video-text retrieval), which raises concern on its effectiveness across a broader range of tasks. Besides, even for video-text retrieval, notable benchmarks such as [YouCook2](https://www.aaai.org/ocs/index.php/AAAI/AAAI18/paper/viewFile/17344/16367) and [MSVD](https://www.aclweb.org/anthology/P11-1020.pdf) are missing, not even in the related work section. Consider acknowledging these works.

iii) From Tab. 2, we know that the model variant "Cross" works the best. But chances are there, other videos from the same batch might be totally irrelevant to the video-of-interest (more pronounced when the batch is small). Could this potentially hurt the learned representation? Also, any insight on what intuitively makes "Cross" better than other model variants?

iv) The conclusion drawn on Contrastive Loss from this work (page six bottom) is opposite to existing works, that is, triplet loss works much better InfoNCE. Any insight or observation besides these mentioned in the paper would be appreciated.

vi) Consider the relevance of [Prototypical Contrastive Learning](https://arxiv.org/pdf/2005.04966.pdf) to this paper (also alleviate the over-discrimination/"false negative" issue on instance discrimination), please acknowledge the work in the related work section.

Minor comments:

i) Please check if the font style follows ICLR'21 guideline, for example, for the title and references.

ii) The green arrow between the positive pair is missing in Fig. 1b.

=========================== Post-Rebuttal ===========================

My questions are well addressed in the rebuttal. I acknowledge the novelty in the paper and the convincing empirical results. The authors have committed to include more datasets to enhance the result completeness and add relevant but missing related work. Taking all these into account, I am adjusting my final rating to Accept.

---

> ### Author Response · Authors · 2020-11-14
> **Response to Reviewer 2**
>
> Thank you R2 for your comments and feedback. We appreciate that you find our paper clearly written, easy to follow and having impressive empirical results. We address your comments about the motivation of our support-set idea below.
>
> **i) The motivation behind using a support set is still unclear, besides the obvious reason of empirical gain. Compare to related techniques such as semi-hard negative mining, what are their relations and the advantage of the proposed approach? Also, since the idea is generic to the training objective, what is the rationale for not using it for the contrastive objective (but instead on the more complicated captioning objective)? Or if you have made attempts, what are the observations?**
> As explained in Section 3.1, the motivation for our additional loss is to address a shortcoming of contrastive learning and the fact that it may use "incorrect" negatives (being unaware of the semantics of the data). The point of our loss is in fact to pull together samples that share a similar caption and that are thus likely to share also similar semantics even though the contrastive loss would like to (incorrectly) push them apart. This is also why our formulation requires the captions -- an extra modality with good semantic information is needed to achieve this relation. So, while applying this idea to a pure contrastive objective is a good goal for future work, the current formulation does not allow it.
>
> **ii) The overall scope of the proposed method is somewhat overstated. Despite that the method is generic to learn video-text representation, the downstream tasks only involve a single type of problem (i.e., video-text retrieval), which raises concern on its effectiveness across a broader range of tasks. Besides, even for video-text retrieval, notable benchmarks such as YouCook2 and MSVD are missing, not even in the related work section. Consider acknowledging these works.**
> We focus on video-text *and* text-video retrieval in the experiments because, since these are the two modalities used for training, we expect the biggest gains there and these are important problems in their own right. We welcome the feedback and added citations to MSVD and YouCook2 to the related work section. In addition, we will add experimental results of our method applied to the MSVD dataset in the revised paper.
>
> **iii) From Tab. 2, we know that the model variant "Cross" works the best. But chances are there, other videos from the same batch might be totally irrelevant to the video-of-interest (more pronounced when the batch is small). Could this potentially hurt the learned representation? Also, any insight on what intuitively makes "Cross" better than other model variants?**
> As shown in Table 1e, there is indeed a negative effect if the batch-size is too small and does not contain relevant samples. Our method performs best when the support-set is medium-sized, thus acting as a bottleneck. We provide an intuition of why "Cross" is best in Table 2 at the bottom of Page 6: this forces videos to "caption each other", thus pulling them together more stringently --- in other words, this is the version of the method that is most extreme in applying our regularization and it is also the one that works the best, indicating that our hypothesis is likely correct.
>
>
> **iv) The conclusion drawn on Contrastive Loss from this work (page six bottom) is opposite to existing works, that is, triplet loss works much better InfoNCE. Any insight or observation besides these mentioned in the paper would be appreciated.**
> A possible reason is that we are testing retrieval performance where triplet-loss training is usually optimal even in the non-video setting (as opposed to, say, classification, where InfoNCE can be expected to be better).
> However, we are now rerunning the InfoNCE experiments trying to optimize them further to check if it is indeed possible to bring their performance closer to that of the triplet loss.
>
> **v) Consider the relevance of Prototypical Contrastive Learning to this paper (also alleviate the over-discrimination/"false negative" issue on instance discrimination), please acknowledge the work in the related work section.**
> We agree that the clustering can also alleviate the strictness of the instance discrimination task, and will update the manuscript to include PCL and other recent relevant works (SwAV) to our related works section.
>
> **Please check if the font style follows ICLR'21 guideline, for example, for the title and references.**
> Thank you for pointing this out, we will update the manuscript.
>
> **The green arrow between the positive pair is missing in Fig. 1b.**
> Thank you for finding this! We will fix this for the revised version.

---

> > ### Comment · AnonReviewer2 · 2020-11-20
> > **Clarification helps; a follow-up question**
> >
> > Thanks for the clarification. For (i), the following clears up my confusion: "The point of our loss is in fact to pull together samples that share a similar caption [...] contrastive loss would like to (incorrectly) push them apart". For (iii), "[...] pulling them together more stringently" makes a lot of sense to me now. Using a captioning task to "weakly" attend to the relevant videos and therefore refine the feature representation is quite ingenious and has shown to be effective (Tab. 2).
> >
> > Besides, just out of curiosity, have you tried using the captioning objective alone (without the contrastive objective)? Intuitively, it should still work for most of the model variants except for "Cross". Any observations or comments will be appreciated.

---

> > > ### Author Response · Authors · 2020-11-20
> > > **Captioning Objective Only Follow-up Response**
> > >
> > > We are glad we were able to clarify the motivation behind our combined objective! Regarding the idea of solely using a captioning objective for training:
> > >
> > > A captioning objective alone does not explicitly align video and text modality embeddings so relying only on a captioning objective actually hurts performance. We did try this, and struggled with getting anywhere near our results that includes a contrastive loss. Our findings would also explain why VidTranslate (Korbar et al. 2020), which solely used only a captioning objective, does not work as well as ours for video-text retrieval even though they used a much bigger architecture than we do (T5-large). However, as an auxiliary task combined with contrastive learning, cross-captionining works really well to combat strictness of instance discrimination contrastive objective.

---

> > > > ### Comment · AnonReviewer2 · 2020-11-20
> > > > **Leaning towards Accept**
> > > >
> > > > Thanks for the clarification. I also noticed it in the Related Work now.
> > > >
> > > > Taking all these into account, I am leaning towards Accept. I will revise my rating.

---

### Author Response · Authors · 2020-11-21
**Revised Version: more experiments/ablations**

We thank all reviewers for their constructive feedback and insightful comments.
At this stage, we have finished our revised version of the paper that we believe has addressed all comments raised and now also includes:

1. **T5 decoder ablation** (as ideated by R3) in Table 1d. A smaller decoder -- with 6 layers instead of 12 -- only decreases performance by 1% (compared to ~3% drop when using a smaller encoder). This likely highlights the regularizing aspect of our captioning task, for which highly accurate captioning is not needed.
2. **Effect of much larger support-set size** (R1), in Table 1f. Our bottleneck hypothesis holds, as a larger support-set simply descreses performance by making the task too easy.
3. **Zero-Shot numbers** for all datasets in Table 11 (R3, R4).
4. **New results on MSVD dataset** (R2): Video-text, text-video retrieval. We outperform SOTA throughout, e.g. by more than 8% for retrieval @1 when pretrained on HT100M.
5. **Action recognition finetuning** results on datasets on HMDB-51, UCF-101 (R2), see new Table 11.
6. **Statistical significance** values for VATEX. We provide the results of training our model for 3 different times on the VATEX dataset in Appendix table 13. We find that our model is SOTA, with 44.9+-0.2 compared to the next best method (35.1%) for retrieval @1.
7. **Added more references**: Additional datasets (LSMDC, YouCook2, MSVD, DiDeMo) and related references (PCL, SwaV) in the related works section and also added UniVL results to the MSR-VTT Table 3 (R2).
8. **Training Details** such as the $\alpha$ and T values in the contrastive objective added to Appendix (R3).
9. **Fixed formatting** (font type of headings) issue (R2). This is fixed at the first uploaded revision to make diff-checking easier.

---

> ### Comment · AnonReviewer2 · 2020-11-21
> **Clarification**
>
> Thanks for the summary. Just to clarify, are results on Tab. 6.6 with linear evaluation or fine-tuning? It mentioned in the text "We then train a linear classifier [...]", indicating the former. But the numbers are significantly higher than SotA with fine-tuning (e.g., Miech et al., 2020), let alone with linear evaluation. A clarification on this would be appreciated. Also, it would be great if SotA methods are included and compared in Tab. 6.6, being consistent with other tables.

---

> > ### Author Response · Authors · 2020-11-21
> > **Action Recognition Results Clarification**
> >
> > Thanks for the question and we are happy to clarify. Our results are with training only a linear-layer, on features extracted from our fixed backbone with or without a learned transformer-pooling head. The backbone, as we describe in the paper, consists of early fusion of features from two models, R(2+1)-D-34 pre-trained on IG65M (motion features) and Resnet-152 pretrained on Imagenet (appearance features).
> >
> > MIL-NCE (Miech et al. 2020), however, trains a S3D-G visual encoder from scratch on HT100M, which makes these two results not comparable.
> >
> > Note that when using pretrained backbones, such as the ones we use, HMDB and UCF numbers are in the range of 75-84% (see Table 16 of [1]) and 88-98% (see Table 6 of [2]), and therefore these are the numbers that are better to compare against.
> > We can add these comparison numbers to Table 6.6 in an updated version of our paper.
> >
> > [1] Duan, Haodong et al. "Omni-sourced Webly-supervised Learning for Video Recognition", ECCV 2020. https://arxiv.org/abs/2003.13042
> >
> > [2] Tran, Du, et al. "A closer look at spatiotemporal convolutions for action recognition", CVPR 2018

---

> ### Comment · AnonReviewer4 · 2020-11-22
> **Thanks for the update + clarifications**
>
> Thanks for the update.
>
> **Zero-shot numbers**: The Table for zero-shot seems to have a wrong caption (ActivityNet is not mentionned for example). Also is the last row MSRVTT or VTT (the table says VTT but in the main text, it is mentioned MSRVTT and not VTT). Please fix.
>
> **Action recognition results**: is the dimension of the features on which you apply the linear layer also 2560 when you apply the transformer pooling? This should be clarified directly in the paragraph of the Appendix to not create confusion. Also the baseline on UCF seems to be very low for such strong features trained on both ImageNet and IG65M. Is there any explanation for that? For example MILNCE trained solely on noisy data from HowTo100M data with only a S3D architecture (so dimension 1024 gets 82.7%).

---

> > ### Author Response · Authors · 2020-11-23
> > **Clarifications for Reviewer 4**
> >
> > We are happy to clarify your two questions:
> >
> > 1. Regrading the zero-shot table, yes the last row should be MSR-VTT and the title should include ActivityNet. We have fixed these two typos in the updated version of the paper.
> >
> > 2. Regarding action recognition results, agreed, our initial reported results on UCF-101 were quite low, and, as we have since realized, was due to not using the right learning rate schedule when training a learning classifier on extracted features. We have updated the manuscript with our updated training schedule, updated numbers on both UCF-101 and HMDB-51, comparison to the state-of-art (as R2 requested), and dimension of transformer pooling representation (1024). We hope that this section is much clearer, but please let us know if anything is still unclear.

---

### Author Response · Authors · 2020-11-23
**Second revised version with improved action recognition section**

We have updated the manuscript with an updated action recognition section:
1. Description of updated training schedule
2. Improved numbers on both UCF-101 and HMDB-51 action recognition with updated training schedule
3. Dimension of transformer pooling representation
4. Comparisons to other state-of-the-art works

---

### Decision · Program_Chairs · 2021-01-07
**Final Decision**

**Decision:**

Accept (Spotlight)

**Comment:**

This paper studies the problem of learning better video-text representation learning with an application to video-text retrieval. It proposes a key innovation: it uses a new generative task of cross-captioning that addresses issues with contrastive learning by learning to reconstruct a sample’s text representation as a weighted combination of a video support set, using a novel objective function using video-set bottlenecks. It uses pre-training based on YouTube video-ASR pairs, and shows empirical results where the proposed method outperforms multiple SOTA methods.

The authors have addressed the feedback of the reviewers, especially with the following improvements:

- Experiments were run on more datasets
- Relevant work pointed out by the reviewers were added
- Concerns regarding technical details were clarified